# The Clinical, Pathological, and Prognostic Value of High PD-1 Expression and the Presence of Epstein–Barr Virus Reactivation in Patients with Laryngeal Cancer

**DOI:** 10.3390/cancers14030480

**Published:** 2022-01-18

**Authors:** Janusz Klatka, Anna Szkatuła-Łupina, Anna Hymos, Maria Klatka, Paulina Mertowska, Sebastian Mertowski, Ewelina Grywalska, Małgorzata Charytanowicz, Anna Błażewicz, Agata Poniewierska-Baran, Dominika Bębnowska, Paulina Niedźwiedzka-Rystwej

**Affiliations:** 1Department of Otolaryngology and Laryngological Oncology, Medical University of Lublin, Jaczewskiego 8 St., 20-954 Lublin, Poland; janusz.klatka@wp.pl (J.K.); annaszkatula@op.pl (A.S.-Ł.); 2Department of Experimental Immunology, Medical University of Lublin, Chodźki 4a St., 20-093 Lublin, Poland; annahymos@gmail.com (A.H.); paulina.lipa56@gmail.com (P.M.); mertowskisebastian@gmail.com (S.M.); ewelina.grywalska@gmail.com (E.G.); 3Department of Pediatric Endocrinology and Diabetology, Medical University, Gębali 1 St., 20-093 Lublin, Poland; mariaklatka@wp.pl; 4Department of Computer Science, Faculty of Electrical Engineering and Computer Science, Lublin University of Technology, Nadbystrzycka 38D, 20-618 Lublin, Poland; mchmat@ibspan.waw.pl; 5Systems Research Institute, Polish Academy of Sciences, Newelska 6, 01-447 Warsaw, Poland; 6Department of Pathobiochemistry and Interdisciplinary Applications of Ion Chromatography, Medical University of Lublin, 1 Chodzki St., 20-093 Lublin, Poland; anna.blazewicz@umlub.pl; 7Institute of Biology, University of Szczecin, Felczaka 3c, 71-412 Szczecin, Poland; agata.poniewierska-baran@usz.edu.pl (A.P.-B.); dominika.bebnowska@usz.edu.pl (D.B.)

**Keywords:** immunophenotype, Epstein–Barr Virus, CD25+, CD69+, PD-1, PD-L1

## Abstract

**Simple Summary:**

Our immune reaction depends on some ‘immune checkpoints’, such as PD-1, PD-L1 and CTLA4, that maintain homeostasis and define new pathways in the fight against carcinogenesis. Viral infections, including EBV (Epstein-Barr Virus) are one of the risk factors for laryngeal cancer. The aim of our study was to evaluate the level of PD-1 receptor in blood, tumor and lymph node samples collected from 45 laryngeal cancer patients and 20 healthy volunteers from control group. We detected the presence of EBV molecules in cancer samples and show the relationship between tumor progression and the level of PD-1 receptor. We confirmed, that EBV infection may affect the PD-1/PD-L1 pathway and develop the laryngeal cancer. What is important, the level of PD-1 on CD4+ T cells in lymph nodes increased the risk of death, so it can be an important prognostic factor (marker) for laryngeal cancer patients’ treatment and their prognosis.

**Abstract:**

Due to the development of molecular diagnostic techniques, the latest research in the diagnosis of cancer diseases, including laryngeal cancer, has been focused on the occurrence of specific types of molecular patterns, including markers expressed on cells of the immune system (e.g., PD-1, PD-L1, and CTLA-4), which may be directly or indirectly involved in the development of neoplastic diseases. Laryngeal cancer is one of the diseases that is diagnosed more often in men than in women, and many factors are involved in its development, including environmental and lifestyle factors, viral infections (e.g., HPV, HHV-1, and EBV), and disorders of the immune system. In this study, we determined the level of PD-1 receptor expression on T and B lymphocytes and their relationships based on the classification of the grade and TNM scale, in turn based on blood, tumor, and lymph node samples from patients diagnosed with laryngeal cancer. In addition, we determined the presence of EBV genetic material in the tested biological materials as well as the degree of cancer advancement and its correlation with the level of PD-1 receptor expression. The results suggested that the level of PD-1 expression on T and B lymphocytes was significantly higher in the tumor samples as compared to the lymph node samples, and their comparison with the immunophenotype results from the blood samples provided statistically significant data on changes in the incidence of individual subpopulations of T and B lymphocytes and the level of PD-1 receptor expression. The analysis of the individual parameters of the TNM scale also showed significant changes between the PD-1 expression and the tested biological material in individual subgroups of the scale. We also found that the expression of PD-1 on the CD4+ T cells from the lymph node samples caused an almost 1.5-fold increase in the risk of death. In the analyses of the presence of EBV, the highest concentration was recorded in the tumor samples, then for the lymph node samples, and followed by the blood samples. Furthermore, we showed that the presence of EBV genetic material was positively correlated with the level of PD-1 expression in the tested biological materials.

## 1. Introduction

From year to year, we observe an increase in the number of people diagnosed with cancer of various origins and stages. Although people diagnosed with laryngeal cancer are not among the world’s top cancers in terms of the number of new cases detected (according to the World Health Organization (WHO) in 2020, it was in the 20th position), it is important to understand this disease and its risks to the population. Statistical data compiled by WHO showed that in 2020, 184,615 new cases of laryngeal cancer were diagnosed worldwide, and over 21% of those were in Europe. Of these, 87.68% were men, and only 12.32% were women, which shows a clear prevalence in one sex. This also applied to the number of deaths, where data showed that in 2020, 19,604 people in Europe died of laryngeal cancer (which accounted for 19.63% of all deaths in the world), 89.87% of whom were men [1]. Due to the development of molecular biology techniques, the latest research on the diagnosis of cancer diseases, including laryngeal cancer, has been focused on the occurrence of a specific type of molecular pattern. The molecular patterns detected in the development of head and neck cancers, including laryngeal cancer, include molecules such as epidermal growth factor receptor (EGFR) [2,3,4] telomerase activity [5], overexpression and amplification of cyclin D1 genes (CCND1) [6], cathepsin D [7], type II estrogen binding sites (EBS) [8], S100-A2 Ca2+ binding proteins [9], type 2 cyclo-oxygenase (Cox-2) [10,11], and galectin-3 [12,13]. As reported in the literature, many scientists have also noted the participation of some viruses in the development of laryngeal cancer, including infection with oncogenic human papillomavirus (HPV), human alphaherpesvirus 1 (HHV-1) [14], and Epstein–Barr Virus (EBV) [15]. Their involvement in the patient’s response to chemotherapy and their correlation to survival time has also recently been studied [16,17,18]. Immunological testing has proven beneficial by determining the immunophenotype involved in the development of laryngeal cancer, which can then provide important data regarding the influence of the immune system on the occurrence and progression of the disease [19]. The immune response is influenced by immunological checkpoints, such as programmed death receptor 1 (PD-1), programmed death-ligand 1 (PD-L1), and cytotoxic T cell antigen 4 (CTLA4), which are key elements in maintaining immune homeostasis and provide information on the occurrence of new pathways in the process of carcinogenesis [20].

In recent years, the research on the PD-1 receptor, which is expressed in numerous immune cells including T-lymphocytes, B-lymphocytes, monocytes, and macrophages has gained prominence. After binding to PD-L1 and PD-L2 ligands, this receptor inhibits the activation of the immune system [21,22] in the regulation of the central and peripheral tolerances and mitigates the antitumor response in the human body. The PD-1/PD-L1 complex inhibits autoimmune responses by blocking T cell proliferation and cytokine production [23]. Some cytokines, such as TGF-β and IL-10 produced by the neoplastic cells, increase the expression of the PD-1 receptor on the lymphocytes, which increases their susceptibility to suppression and even apoptosis [24,25,26]. Therefore, scientists have concluded that the PD-1/PD-L1 pathway is one of the mechanisms by which cancer cells escape immune control. Further research into the role of immune checkpoints may lead to the development of new treatment options targeting the tumor’s response as opposed to the tumor itself [20].

Therefore, in our research, we decided to investigate two key issues that may be involved in the development of laryngeal cancer. The first was whether the expression of the PD-1 receptor on the cells of the immune system in patients diagnosed with laryngeal cancer influences disease progression (based on the TMN scale) and could be one of the prognostic markers. For this purpose, we analyzed the level of PD-1 receptor expression on T and B lymphocytes in peripheral blood, tumor, and lymph node samples, which then allowed us to determine the degree of differentiation in the expression of the tested receptor. The second issue was to determine the presence of EBV genetic material in the tested biological material while considering the advancement grade of the cancer and its correlation with the level of PD-1 receptor expression.

## 2. Materials and Methods

### 2.1. Patients and Healthy Group

Forty-five untreated patients with laryngeal squamous cell carcinoma (LSCC) were enrolled in the Department of Otolaryngology and Oncology of the Medical University of Lublin. They were between 50 and 79 years of age, with a mean age of 62.27 ± 6.40 years. Each patient’s medical status was confirmed by histological diagnosis. Twenty volunteers hospitalized due to a distortion of the nasal septum were enrolled in the control group. The control group were anti-VCA IgM-negative and anti-VCA IgG-positive. They were between 44 and 69 years of age, with an average age of 58.45 ± 7.03 years. The main clinical features of the study group are detailed in Table 1. The clinical and histology stages of laryngeal cancer patients were classified based on the TNM staging system and histological grading (G1–G3). The study was conducted according to the guidelines of the Declaration of Helsinki and approved by the Ethics Committee of the Medical University of Lublin (KE-0254/70/2015, approval date: 26 March 2015).

Patients with laryngeal cancer and the control group tested negative for human immunodeficiency virus, hepatitis B virus, hepatitis C virus infection, and allergic diseases. The participants in this study were not taking any medications that could affect the immune system, nor had any undergone recent blood transfusions.

### 2.2. Flow Cytometric Analysis of CD69+, CD25+, and PD-1 Cells in Peripheral Blood

Peripheral blood samples for the frequency analysis were collected into tubes containing EDTA. Cells were stained for 30 min at 4 °C in the dark with the following surface antibodies: mouse anti-human antibodies fluorescein isothiocyanate (FITC)-CD3, phycoerythrin (PE) anti-CD19, FITC anti-CD4, PE anti-CD8, phycoerythrin-cy5 (PE-cy5) anti-CD69, PE-cy5 anti-CD25, FITC anti-CD19, FITC anti-CD4, FITC anti-CD8, and PE anti-PD-1.

Following staining of samples with antibodies, cells were treated with lysing solution (Becton Dickinson, East Rutherfort, NJ, USA) and incubated for 10 min in 4 °C in the dark. Afterwards, the cells were washed twice with phosphate-buffered saline (PBS). All the antibodies and their corresponding isotype controls were purchased from Becton Dickinson, East Rutherfort, NJ, USA. Lymphocyte populations were defined by forward-scattering light (FSC) and side-scattering light (SSC) adjustments. The cell subsets were detected by different cell labeling and gating. The data were collected on a four-color FACSCalibur flow cytometer (Becton Dickinson, East Rutherfort, NJ, USA). CellQuest software was used for data analysis, and the percentage of positive cells was recorded. At least 10,000 events were acquired from each sample.

### 2.3. Flow Cytometric Analysis of CD69+, CD25+, and PD-1 Cells in Tumor Tissue and Lymph Nodes

Tumor samples and lymph nodes were obtained during surgical treatment. Tissues were digested with 1 mg/mL of type I collagenase (Sigma-Aldrich, Steinheim, Germany). The resulting cell suspensions were filtered through a 40 μm nylon cell strainer (Falcon; Becton Dickinson, East Rutherfort, NJ, USA) to removed large particles. Then, cell suspensions were used for the separation of peripheral blood mononuclear cells (PBMCs) by density gradient centrifugation using Gradisol-L (Aqua Medica, Łódź, Poland) (700 G for 20 min at room temperature). Interphase cells were collected and washed twice in PBS without Ca^2+^ or Mg^2+^ (Biochrom AG, Berlin, Germany). The PBMC were resuspended at a density of 2 × 106 cells/mL in PBS and were stained with the following for 30 min at 4 °C in the dark: mouse anti-human antibodies FITC anti-CD3, PE anti-CD19, FITC anti-CD4, PE anti-CD8, PE-cy5 anti-CD69, PE-cy5 anti-CD25, FITC anti-CD19, FITC anti-CD4, FITC anti-CD8, and PE anti-PD-1. Following staining of samples with antibodies, the cells were washed twice with PBS. Stained cells were subjected to cell phenotype characterization using a flow cytometric analysis. Figure 1, Figure 2 and Figure 3 show an example of the cytometric analysis of PD-1 lymphocytes in blood, tumor tissue, and lymph node.

### 2.4. DNA Isolation, RT-PCR, and Calculation of EBV Load

DNA was isolated from five million PBMCs using the QIAamp DNA Blood Mini Kit (QIAGEN, Hilden, Germany) according to the manufacturer’s instructions. The concentration and purity of the isolated DNA were verified with a BioSpec-nano spectrophotometer (Shimadzu, Kyoto, Japan). All the samples were analyzed in duplicate with standard calibrators and a negative control (DNA elution buffer) included. The ISEX version of the EBV PCR kit (GeneProof, Brno, Czech Republic) was used to calculate the number of EBV-specific DNA copies. The PCR was conducted using the 7300 Real-Time PCR System (Applied Biosystems, Foster City, CA, USA). The detection threshold of the assay was 10 EBV DNA copies per μL.

### 2.5. Evaluation of IgM VCA and IgG VCA Concentrations

A plasma sample was taken from all participants for the detection of IgM VCA and IgG VCA antibodies by commercial immunoenzymatic assay kits (ELISA) and stored at −80 °C before analysis. ELISA assay was performed according to the manufacturer’s instructions for the detection of IgM VCA and IgG VCA levels (Demeditec, Germany, with a limit of detection of 10 U/mL). The OD value at 450 nm was measured. The concentration of anti-VCA IgM and anti-VCA IgG were calculated according to the standard curve.

### 2.6. Statistical Analysis

The continuous experimental data were expressed as mean ± SD, median, and range and analyzed by Statistica 12 software (StatSoft, Kraków, Poland). Several types of statistical analyses were performed. Normality of data distribution was tested with the Shapiro–Wilk test. Levene’s test was used to evaluate the homogeneity of variance. Independent or paired Student’s *t*-tests were used when comparing normally distributed variables, and the Mann–Whitney U test and Wilcoxon signed rank test were used in cases of non-normally distributed variables. The differences between more than two groups were analyzed by ANOVA or ANOVA Kruskal–Wallis. Pearson’s correlation coefficients and corresponding significance tests were used to assess the strength and direction of the linear relationships between pairs of variables. The Cox proportional hazard regression model was also evaluated. A value of *p* < 0.05 was considered statistically significant.

## 3. Results

### 3.1. Characteristics of the Immunophenotype in Patients Diagnosed with Cancer of the Larynx and in the Control Group

The study group consisted of 45 patients diagnosed with laryngeal cancer at various stages whose mean age was 62 years (62.27 ± 6.40). The control group consisted of 20 age-matched subjects (mean age 58.450 ± 7.03 years). In the first phase of the study, the immunophenotype characteristics of both tested groups were described, which enabled the determination of the frequency of the individual lymphocyte subpopulations in the peripheral blood samples, such as the presence of T lymphocytes (CD3+), B lymphocytes (CD3+ CD19+), T helper lymphocytes (Th) (CD3+ CD4+), cytotoxic T lymphocytes (Tc) (CD3+ CD8+), regulatory lymphocytes (Treg) (CD4+ CD25+; CD8+ CD25+), and CD19+ CD25+ on B cells and a subpopulation of activated lymphocytes (CD4+ CD69+, CD8+ CD69+ T cells, and CD19+ CD69+ B cells). From these results, we found that among patients diagnosed with laryngeal cancer, there was a significantly higher percentage of CD19+ B cells (1.42×) and Treg (1.52× for CD3+ CD25+ T cells, 1.29× for CD4+ CD25+ T cells, and 3.39× for CD8+ CD25+ T cells), as well as for CD3+ CD69+ T cells (1.90×), CD4+ CD69+ T cells (2.20×), and CD8+ CD69+ T cells (3.38×), as compared to the control group (Table 2).

In addition, we found a higher level of PD-1 receptor expression on the CD4+ T cells (2.66-fold increase) and the CD8+ T cells, the level of which was 4.35 times higher than in the control group. In the control group, statistically significant higher mean values were recorded for the presence of CD3+ T lymphocytes and CD3+ CD4+ T lymphocytes (1.07 times and 1.11 times, respectively), as well as a PD-1 expression level that was 58.41% higher in CD19+ B lymphocytes (Table 2).

We did not observe statistically significant changes in NK cells levels; CD3+ CD8+ T cells; or the ratio of CD3+ CD4+ T cells to CD3+ CD8+ T cells, CD19+ CD25+, or CD19+ CD69+ B cells in either study group (Table 2).

### 3.2. Characterization of the Immunophenotype of Tumor and Lymph Node Samples in Patients Diagnosed with Laryngeal Cancer

Due to the demonstration of numerous statistically significant changes in the immunophenotype of patients diagnosed with laryngeal cancer in relation to the control group, we decided to take a closer look at the changes that occurred in the individual subpopulations of T and B lymphocytes in samples obtained from the tumor and the lymph nodes. We also analyzed the expression of PD-1 in the samples. All the results obtained from these experiments were also subjected to a comparative analysis with the results obtained from the blood samples. The conducted analysis showed that, within the immunophenotype determined from the tumor samples of patients with laryngeal cancer, there were statistically significant differences in the levels of occurrence of individual lymphocyte subpopulations (Table 3).

These changes concerned, in particular, increases in the presence of CD3+ CD69+ T cells (by a factor of 1.51), CD4+ CD69+ T cells (by a factor of 3.42), CD8+ CD69+ T cells (by a factor of 2.50), and CD8+ CD25+ T cells (by a factor of 1.37) in the tumor vs. the lymph node samples. We also found that the level of PD-1 expression on the T and B cells was significantly higher in the tumor samples, as compared to the lymph node samples, by a factor of 2.44 for CD19+ B cells, 1.76 for CD8+ T cells, and 1.94 for CD4+ T cells. A comparison of the immunophenotype results from the tumor and lymph node samples to the blood analysis provided statistically significant differences in the frequency of the sub-populations of T and B lymphocytes and the levels of PD-1 receptor expression.

When the immunophenotypes of the tumor samples were compared with the blood samples, statistically significant differences in the incidences of the CD3+ CD69+, CD4+ CD69+, and CD8+ CD69+ T cells in the tumor samples were 7.83, 9.67, and 10.5 times higher, respectively. In addition, we also recorded a 3.79 times higher level of CD19+ CD69+ B cells in the tumor samples. The opposite tendency was observed in the frequency of CD3+ CD25+ and CD4+ CD25+ T cells, which were 1.57 and 2.11 times higher, respectively, in the blood samples (Table 3).

In all analyzed tumor samples, the level of PD-1 expression in T and B lymphocytes was significantly higher in the tumor samples as compared to the blood samples. The biggest difference was observed for B lymphocytes, where the expression levels in the tumor samples were over five times higher (5.41 times), over two times higher for CD8+ T cells (2.34 times), and almost two times higher for CD4+ T cells (1.79 times) (Table 3).

In the analysis of the frequency of individual lymphocyte subpopulations between the lymph node and blood samples, significant changes were also found in the CD69+ markers, the level of which was significantly higher in the lymph nodes. The greatest changes were noted for the CD3+ T cells, the increase in which was more than 5.18 times higher in the lymph nodes as compared to the CD19+ CD69+ B (4.31 times), CD8+ CD69+ T (4.21 times), and CD4+ CD69+ T cells (2.82 times). In the lymph nodes, we observed a higher level in the CD25+ marker in the CD3+ T (1.63 times), CD4+ T (1.80 times), and CD19+ B cells (1.42 times) as compared to the blood samples. In the analysis of the PD-1 expression levels, a statistically significant difference in the lymph node samples was demonstrated for CD19+ B cells, the level of which was more than 2.22 times higher than the results from the blood samples (Table 3). A significantly higher percentage of the T and B lymphocytes positive for PD-1 antigen expression in the laryngeal cancer tissue, as compared to their percentages in the blood and lymph nodes, indicated that the anti-tumor defense mechanisms were particularly weakened in the neoplastic tissue.

### 3.3. Analysis of the Relationship among the Expression of the PD-1 Receptor in the Tested Blood, Tumor, and Lymph Node Samples of Patients Diagnosed with Laryngeal Cancer and the Stage of the Disease and the TNM Scale

The aim of the next analysis was to determine the relationship between the expression of the PD-1 receptor on the T and B lymphocytes and the progression of laryngeal cancer. For this purpose, the grade of laryngeal cancer and the TNM scale were used for analysis, where T was the size of the tumor, N was the involvement of the local lymph nodes, and M was the presence of metastases on distant organs. The correlation was performed considering the individual grades of all analyzed grade parameters (G1–G3) of the TNM scale (T2–T4; N0–N3; and M0–M1).

From the results, we found that the level of PD-1 receptor expression on the individual cells of the immune system showed significant changes in the analyzed parameters, which indicated the stage of laryngeal cancer, as well as in the biological material tested. In the analysis of PD-1 expression based on the degree of disease progression (G), we found significant differences only for T lymphocytes among all the samples. In the peripheral blood samples, there were only significant differences in CD8+ PD-1 T cells between G1 and G3 (1.46×), as well as between G2 and G3 (1.64×) (Table 4). In the tumor and lymph node samples, we observed an upward trend in the PD-1 receptor expression on TCD4+ and CD8+ lymphocytes with an increase in the grade of the disease (Table 4). The results suggested that the level of PD-1 receptor expression may be related to the stage of the laryngeal cancer, based on the results from the tumor and lymph node samples as compared to the blood samples.

Another analyzed parameter was the determination of the relationship between PD-1 expression and tumor size (T). From the results, as presented in Table 5, we concluded that only CD8+ PD-1 T cells in the peripheral blood samples show statistical differences between the T2 and T4 groups and the T3 and T4 groups. This indicated that the expression of the PD-1 receptor on the cells of the immune system does not depend on the tumor size in patients diagnosed with laryngeal cancer.

The analysis of PD-1 expression in the presence of metastases in local lymph nodes (N) provided the most information. We have shown that statistically significant differences in the expression level of this immune checkpoint occurred in both T and B lymphocytes in the peripheral blood and tumor samples as compared to the lymph node samples, where the differences were only in the T lymphocyte subpopulation (Table 6). A detailed analysis of the individual groups showed that the PD-1 expression on the B lymphocytes and both the T lymphocyte subpopulations was higher in the tumor samples as compared to the peripheral blood samples in all groups of the N parameter. In addition, we showed statistically significant differences in the level of PD-1 receptor expression in the individual N = 0–N = 3 groups in each type of biological material, which are presented in Table 6. Particularly noteworthy was that the expression level of the PD-1 receptor on the T CD4+ and T CD8 + lymphocytes from the tumor and lymph node samples increased with the N scale (more precisely, from N = 1 to N = 3). The results suggested that the level of PD-1 expression may be related to the degree of lymph node involvement.

The last analyzed parameter was the presence of metastases in distant organs, which was expressed as the M parameter. From the results, we concluded that the expression of the PD-1 receptor on the CD8+ T lymphocytes in all the tested biological materials, as well as the CD4+ T in the tumor and lymph node samples, significantly correlated with the occurrences of metastases in distant organs (Table 7).

### 3.4. Analysis of PD-1 Expression in the Studied Biological Material Regarding Life Expectancy of Patients Diagnosed with Laryngeal Cancer Based on the Cox Proportional Hazard Regression Model

Due to significant differences in the levels of PD-1 expression in the studied materials, we decided to analyze which, if any, were significant for the survival of patients diagnosed with laryngeal cancer. For this purpose, we used Cox’s hazard proportional regression analysis. The results shown in Table 8 suggested that at least one independent variable was significantly related to the patient survival rate. This applied to the PD-1 expression on the CD4+ T cells from the lymph node samples, which caused an almost 1.5-fold increase in the risk of death in the participants. The Cox proportional hazard analysis suggested that the evaluation of PD-1 expression on CD4+ T cells located in the lymph nodes may be a prognostic factor.

### 3.5. Analysis of the Presence of EBV in the Tested Biological Materials and Related to the Grade (G) and TNM Scale of Laryngeal Cancer Classification

The second important aspect related to the pathogenesis of laryngeal cancer was to establish the presence of the EBV virus in the tested biological material and its role in disease progression based on the stage and TNM scale.

The analyses showed that in 48.9% of the tested patients, EBV was detected in the biological matter. Its highest concentration (based on the mean amount of EBV DNA copy number/µg DNA) was recorded for the tumor samples and then for the lymph node samples, and the lowest concentration was shown in the blood samples. This was especially true between G = 2 and G = 3 (Table 9). The observed increase in EBV copy number was a factor of 3.48× x for the lymph node samples and 4.22× for the peripheral blood samples.

In the next analyzed parameter, the tumor size (T), we observed a similar trend as had been found in the G parameter. We noted an increase in the number of EBV virus copies in the lymph node samples by a factor of 3.36× between T = 3 and T = 4, as well as a factor of 3.58× in the peripheral blood samples also between T = 3 and T = 4 (Table 10).

We did not observe statistically significant changes in the relationship between the amount of EBV and the N and M parameters associated with metastases and the local lymphatic nodes and distant organs (Table 11 and Table 12). Additionally, quantifications of IgM and VCA EBV IgG were performed, which showed that the amounts of VCA EBV IgM were statistically significantly dependent on the N parameter (Table 11). The level of IgM VCA EBV increased with the increase in the N groups (from N = 1 to N = 3). The observed change in the concentration of IgM VCA EBV between the groups N = 1 and N = 3 was 2.89 times greater, while between N = 2 and N = 3, it increased 1.83-fold.

### 3.6. Analysis of the Correlation between the Presence of EBV Genetic Material and the Expression of PD-1 in the Tested Biological Material from Patients Diagnosed with Laryngeal Cancer

Due to the statistically significant results of the amount of EBV genetic material in the tested samples obtained from previous analyses, we analyzed the correlation of these results with the levels of PD-1 receptor expression. The analyses show that the presence of EBV genetic material positively correlated with the level of PD-1 expression in the tested biological materials. We noted a statistically significant correlation between EBV DNA copy number/µg DNA and PD-1 expression on the CD8+ T cells in the blood samples as well as on the CD4+ and CD8+ T cells in the tumor and lymph node samples (Table 13). We found the same tendency in the amount of EBV genetic material detected in the tumor and lymph node samples, as well as in the level of PD-1 expression on CD4+ and CD8+ T cells in all the samples (Table 13). The positive correlation of the presence of EBV genetic material to the percentage of lymphocytes expressing PD-1 (i.e., CD4+ T lymphocytes in the blood and CD4+ and T CD8+ T cells in laryngeal cancer tissue and lymph nodes) may indicate a negative role for EBV in the process of anti-cancer defense.

## 4. Discussion

### 4.1. Immunophenotype of Patients Diagnosed with Cancer of the Larynx

In studies of the immunophenotypes in patients diagnosed with laryngeal cancer, the changes in the number of individual lymphocyte subpopulations were observed in relation to a control group, with the differences being related to an increase in CD4+ CD3+ lymphocytes and a decrease in CD19+ B and CD3+ T lymphocytes as compared to the control group, which suggested that they have a significant role in the pathogenesis of laryngeal cancer. Studies conducted by other researchers have also shown a reduced absolute number of CD3+, CD4+, and CD8+ T cells in patients with laryngeal cancer as compared to the control group [27]. However, the studies conducted by Chen’s team found that the total number of lymphocytes was significantly reduced as compared to the control group, and the reduction in CD4+ T cells was not statistically significant [28]. Our studies to date on the differences in the percentage of T lymphocytes have indicated that the number of these lymphocytes largely depended on tumor resection [29]. Additionally, the results related to Treg lymphocytes, which may influence the regulation of the immune response in the cancer environment, were noteworthy. In our study, some percentages of Treg cells were activated early (CD69+ T cell) and late (CD25+ T cell). In our study, we not only found a higher percentage of CD69+ and CD25+ T cells in the blood samples as compared to healthy controls, but we also observed an increase in these lymphocytes in the tumor samples as compared to the lymph node samples. An assessment of the initial phase of the immune response to the tumor antigen in the tumor-draining lymph nodes is crucial. The tumor antigen presentation by the dendritic cells to the naive lymphocytes in the lymph nodes plays an important role in the circulation of the antigen-specific lymphocytes. Active anti-tumor lymphocytes are released into circulation and migrate into the tumor tissue [30,31]. Starska et al. demonstrated a statistically significant increase in the expression of CD69 in an experiment with T lymphocyte stimulation by laryngeal cancer cells [32]. The increased frequencies of CD69+ lymphocytes in the tumor microenvironment may be the result of exposure of lymphocytes with tumor antigens. In the analysis of the frequency of individual lymphocyte subpopulations between the lymph node and blood samples, significant changes were also found in the CD69+ marker, the level of which was significantly higher in the lymph node samples. In the lymph node samples, we observed a higher level of the CD25+ marker on the CD3+ T, CD4+ T, and CD19+ B cells. The studies available in the literature concerning the level of CD69+ expression on the T lymphocytes have shown that the level of this marker measured in peripheral blood samples in patients with head and neck cancer was lower in patients with cancer as compared to a control group, but these differences have not been shown to be statistically significant [30,31]. Starska et al. suggested that higher expression of CD69 with low expression of CD25 could indicate the occurrence of dysfunction in the regulatory mechanism in laryngeal cancer [32]. Sheu et al. demonstrated a depressed expression of CD25 with a highly expressed CD69 on lymphocytes in the tumor microenvironment. Due to the downregulation of CD25 on lymphocytes, the immune response in cancer was able be altered and functionally inhibited [33]. In contrast, many studies have confirmed that the percentage and absolute number of CD4+ CD25+ T cells increased in the peripheral circulation of patients diagnosed with malignant neoplasms [32,33,34].

### 4.2. PD-1 Expression on T and B Lymphocytes

More and more research and clinical studies have focused on linking immunological checkpoints in this PD-1/PD-L1 pathway with the prognosis and progression of many neoplastic diseases, including kidney cancer [35], chronic lymphocytic leukemia [36,37], multiple myeloma [38], melanoma [39], breast cancer [40], and lung cancer [41], as well as head and neck cancers [42]. In our study, we showed statistically significant changes in the level of PD-1 receptor expression in the blood, tumor, and lymph node samples from patients diagnosed with laryngeal cancer. Among the blood samples, we found a higher level of PD-1 receptor expression on CD4+ and CD8+ T cells than in the control group. Similar research results were obtained by the Hsu team, who investigated the level of PD-1 receptor expression in nasopharyngeal carcinoma. They showed that higher levels of PD-1 expression on CD8+ T cells resulted in poorer clinical outcome of the tested patients [43]. Additionally, in our analyses, we also showed that the level of PD-1 expression on T and B lymphocytes was significantly higher in the tumor samples as compared to the lymph node samples for the CD19+ B, CD8+ T, and CD4+ T cells, and the level of PD-1 expression on T and B cells was significantly higher in the tumor samples than in the blood samples. Although the literature has been dominated by studies on the level of PD-1 receptor ligand (PD-L1) expression in the development and progression of neoplastic diseases, the PD-1/PD-L1 pathway interrelationships have been shown to be closely related. PD-L1 has been reported to be overexpressed in most cancers, including nasopharyngeal carcinoma, inhibiting the T cell-dependent anti-tumor immunity via PD-1 on TIL [44,45]. Analysis by Smith’s team found that the level of PD-L1 expression tested in tumor samples in Caucasian patients was associated with a poor prognosis in those diagnosed with nasopharyngeal carcinoma [46]. Recent studies have also shown that PD-L1 expression is strongly associated with a poor prognosis in patients with diagnosed neoplasms of epithelial origin [47]. These studies suggested the effect of PD-L1 on the induction of the tumor progression by disrupting anti-tumor immunity [48]. However, the prognostic significance of both PD-L1 and PD-1 in the development of laryngeal cancer and related neoplasms of head and neck cancers has not yet been fully understood. Some studies have suggested an association between high levels of PD-L1 expression in nasopharyngeal carcinoma patients and a shortened survival time [44,49] despite the lack of high PD-1 expression associations [44]. Our analyses showed that the level of the PD-1 receptor expression in the blood samples, depending on the stage of the disease (G), was statistically significant for the CD8+ T cells, while in the tumor and lymph node samples, significant expression of this molecule was found on the CD4+ and CD8+ T cells. The analysis of the individual parameters of the TNM scale also showed significant changes between the PD-1 expression and the tested biological materials in the individual subgroups of the scale. In the T parameter, a statistically significant change was found in the level of PD-1 expression on the CD8+ T cells in the analysis of the blood samples. Regarding the N parameter, we found that the level of PD-1 expression on the T and B lymphocytes in almost all the analyzed blood, tumor, and lymph node samples was statistically significant. The exceptions were the CD4+ T cells from the blood samples and the CD19+ B cells from the lymph node samples. This study also showed that significant differences between the PD-1 expression and the occurrence of metastases to distant organs (M) concerned the CD8+ T cells in the blood samples as well as the CD4+ and CD8+ T cells in the tumor and lymph node samples. We also found that the PD-1 expression on CD4+ T cells from lymph node samples was associated with a 1.5-fold increased risk of death among patients diagnosed with laryngeal cancer. The demonstrated differences in the present study may be related not only to the differences resulting from the types of analyzed neoplasms, but also to the heterogeneous application of experimental procedures, patient samples, and survival endpoints. However, most of the available literature data have shown that a relatively higher expression of PD-L1 or PD-1 may be indicative of a worse prognosis in cancer patients [50,51,52,53,54]. A typical reason for this is that PD-1 and PD-L1 interactions lead to immunosuppression and promote tumor progression [55,56]

### 4.3. The Role of EBV in the Development of Laryngeal Cancer

Estimated data have indicated that nearly 90% of the world’s population may be carriers of EBV, which can remain dormant. However, a growing body of scientific research has indicated that this seemingly innocent virus may be etiologically related to the precancerous lymphoproliferative diseases (LPDs) and numerous different cancers (e.g., Hodgkin’s lymphoma [57], Burkitt lymphoma [58,59], and nasopharyngeal carcinoma [60,61]). Analyses by the Kwok-Fung Lo team on another type of nasopharyngeal carcinoma showed that the presence of EBV genetic material in patient blood had a strong correlation to the tumor stage and overall patient survival [62]. Our analyses also indicated that the presence of EBV genetic material was positively correlated to the level of PD-1 expression in the tested biological materials. We noted a statistically significant correlation between EBV DNA copy number/µg DNA and PD-1 expression on the CD8+ T cells in the blood samples as well as on the CD4+ and CD8+ T cells in the tumor and lymph node samples. We confirmed the same tendency in the amount of detected EBV genetic material in the tumor and lymph node samples and the level of PD-1 expression on the CD4+ and CD8+ T cells in all samples. Our study found that the level of PD-1 receptor expression and the presence of EBV, depending on the type of biological material analyzed, affected the progression of laryngeal cancer based on individual groups of the TNM scale. In addition, along with the increase in PD-1 receptor expression in the analyzed samples, the amount of detected EBV also increased. This agreed with the results of the nasopharyngeal carcinoma study by Fang’s team, where LMP1 was shown to induce PD-L1 cell expression in vitro [63,64]. Additionally, recent research has been focused on anti-PD-1 and anti-PD-L1 therapeutic approaches. The data presented by the Fang team confirmed that blocking the expression of one of the EBV surface proteins LMP1 or PD-L1 could be used as a therapy against nasopharyngeal carcinoma [63,64]. A comprehensive analysis of studies comparing the safety and efficacy of anti-PD-1 monotherapy, chemotherapy, and their combination in the treatment of nasopharyngeal carcinoma was performed in [65]. In general, EBV infection rates in patients with cancer have indicated its contribution to carcinogenesis [57,58,59,60,61]. Viral infection may manipulate the PD-1 expression in immune cells [41,66]. PD-1-positive lymphocytes are a crucial component of immune tolerance. According to our results, an EBV infection would be likely to induce the expression of PD-1 on lymphocytes. However, it is not yet clear whether overexpression may be an exclusive feature of EBV-positive patients with laryngeal cancer. Further study including larger amounts of EBV-positive subjects is needed.

Our study suggested that the immune response to chronic infection may favor the recruitment of PD-1-positive immune cells in tumors, lymph nodes, and peripheral blood. An EBV infection combined with an increase in PD-1+ lymphocytes could be a predictive biomarker for the efficacy of immunotherapy in laryngeal cancer. Our study had limited data due to the lack of PD-1 assessments in cancer tissue. However, Kleinovink et al. indicated that the expression of the checkpoint molecules in infiltrating immune cells may be a more accurate indicator for treatment efficacy [67].

### 4.4. Limitations of Conducted Research

Despite obtaining statistically significant results, our research had some limitations. These include the small sample size of the tested patients diagnosed with laryngeal cancer. Some of the patients recruited for the study did not reflect the entire population of people affected by this disease, which may have also affected the results of our research. In addition, only male patients participated in this study, which limited the generalizability of our results to both sexes. The strength of our research was that it performed a comprehensive analysis of both PD-1 receptor expression and the presence of EBV in three types of biological material (i.e., tumor, lymph node, and peripheral blood samples) and interpreting the results based on the severity of disease and the TNM scale. The results provided information on the contribution of the analyzed parameters to the development and progression of laryngeal cancer. However, further studies are needed to clarify the exact role of the PD-1 receptor and the presence of EBV genetic material in the pathogenesis of laryngeal cancer, as well as to extend the scope to a larger, more diverse population (e.g., female patients). Ongoing research may determine whether the expression of PD-1 receptors on TCD4+ lymphocytes could have prognostic value, which could improve the diagnostic accuracy of laryngeal cancer.

## 5. Conclusions

Our research showed that, among patients diagnosed with cancer of the larynx, there were changes in the occurrence of subpopulations of T and B lymphocytes as compared to the control group, as well as among the analyzed samples of biological materials. These changes also concerned the differences in the level of PD-1 receptor expression between the experimental participants and the control group, as well as their levels within the tumor, lymph node, and blood samples. The differences in the amount of PD-1 receptor expression, as well as in the changes in the percentage of individual lymphocytes in T and B lymphocytes, may be useful in determining disease progression as well as providing information regarding the pathogenesis of laryngeal cancer, including a disturbance in the PD-1/PD-L1 pathway and the T cell activation process.

Our research also suggested that the level of PD-1 receptor expression may be related to the presence of EBV. The presence of EBV, depending on the type of biological material analyzed, affected the progression of laryngeal cancer based on individual groups of the TNM scale. We also showed that, along with an increase in PD-1 receptor expression in the analyzed samples of biological materials, the amount of detected EBV also increased. This indicated that chronic infection with EBV may affect the functioning of the immune system, and the dysregulation of the basic mechanisms of immune reaction can interfere with the effective functioning of the PD-1/PD-L1 pathway, allowing for the development of neoplastic cells in laryngeal cancer. In conclusion, the significantly higher percentage of T and B lymphocytes positive for PD-1 antigen expression in the laryngeal cancer tissue, as compared to their percentage in the blood and lymph node samples, indicated that the neoplastic tissue was a site where defense mechanisms were particularly weakened. Moreover, the positive correlation of the presence of EBV genetic material with the percentage of lymphocytes expressing PD-1 (CD4+ T lymphocytes in the blood and CD4+ and CD8+ T lymphocytes in the laryngeal cancer tissue and lymph nodes) indicated a negative role for EBV in the process of antitumor defense. Based on the Cox proportional hazard analysis, we concluded that the assessment of PD-1 expression on the CD4+ T cells located in the lymph nodes has potential as a prognostic factor.

## Figures and Tables

**Figure 1 cancers-14-00480-f001:**
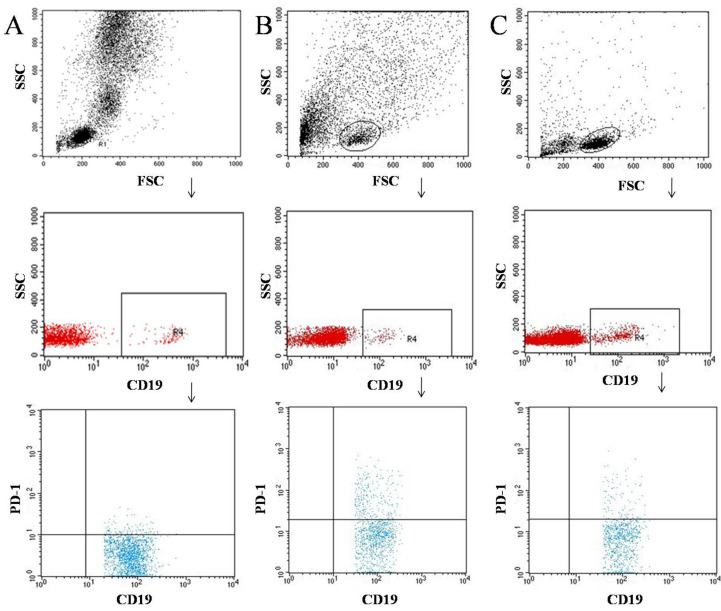
Example of the cytometric analysis of CD19+ PD-1+ lymphocytes in blood sample (**A**), tumor sample (**B**), and lymph node sample (**C**).

**Figure 2 cancers-14-00480-f002:**
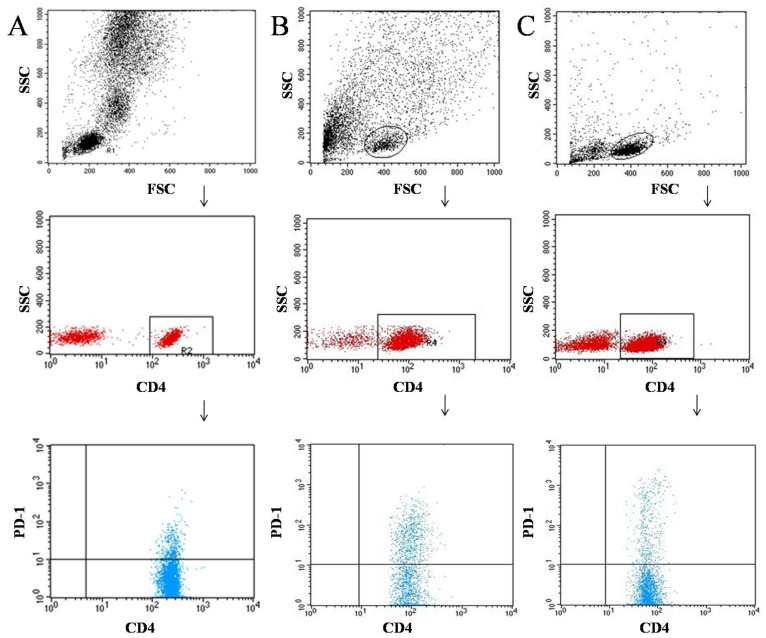
Example of the cytometric analysis of CD4+ PD-1+ lymphocytes in blood sample (**A**), tumor sample (**B**), and lymph node sample (**C**).

**Figure 3 cancers-14-00480-f003:**
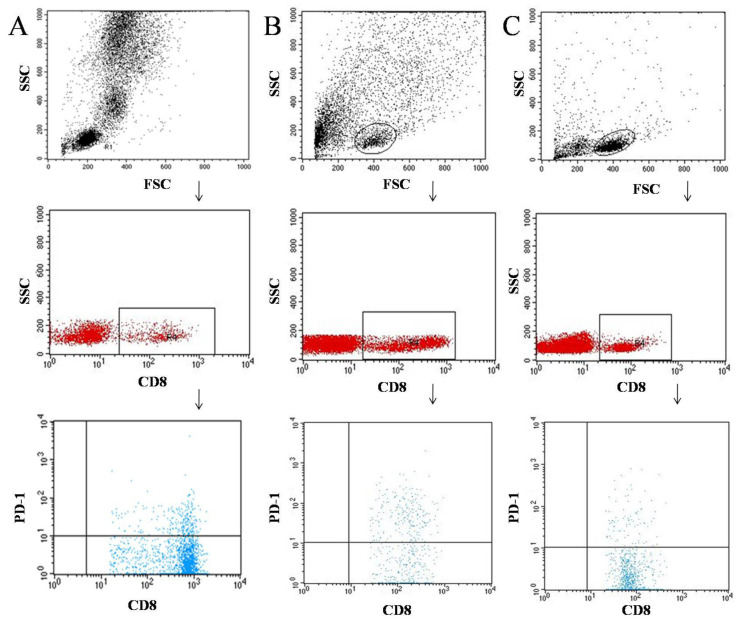
Example of the cytometric analysis of CD8+ PD-1+ lymphocytes in blood sample (**A**), tumor sample (**B**), and lymph node sample (**C**).

**Table 1 cancers-14-00480-t001:** The main clinical features of the study group.

Characteristics	Scale	Laryngeal Cancer Patients (*n* = 45)
Age, Mean ± SD Years		62.27 ± 6.40
Gender (Male/Female)		45/0
Tumor stage	TI	0
	TII	6.7% (3)
	TIII	28.9% (13)
	TIV	64.4% (29)
Nodal stage	N0	8.9% (4)
	NI	22.2% (10)
	NII	57.8% (26)
	NIII	11.1% (5)
M stage	M0	80% (36)
	MI	20% (9)
Grading	GI	26.7% (12)
	GII	37.8% (17)
	GIII	35.5 (16)
Anti-VCA IgM	Positive	49% (22)
	Negative	51% (23)
Anti-VCA IgM (U/mL)	Mean ± SD	25.59 ± 10.36
	Median (min–max)	23.07 (15.31–49.91)
Anti-VCA IgG	Positive	100% (45)
Anti-VCA IgG (U/mL)	Mean ± SD	90.51 ± 51.79
	Median (min–max)	74.36 (26.22–242.62)
EBV DNA	Positive	48.9% (22)
	Negative	51.1% (23)
EBV DNA copy number/µg DNA in the blood	Mean ± SD	259.42 ± 207.96
	Median (min–max)	162.66 (32.47–718.92)
EBV DNA copy number/µg DNA in the tumor tissue	Mean ± SD	582.71 ± 398.91
	Median (min–max)	556.57 (75.76–1542.67)
EBV DNA copy number/µg DNA in the lymph node	Mean ± SD	430.45 ± 295.70
	Median (min–max)	379.75 (59.89–867.91)

**Table 2 cancers-14-00480-t002:** General characteristics of the immunophenotype of patients diagnosed with cancer of the larynx and the control group, according to blood samples.

Characteristic/Frequency of Occurrence [%]	Patients Diagnosed with Laryngeal Cancer (*n* = 45)		Control Group (*n* = 20)		*p* Value
	Median	Range	Median	Range	
Age	63.000	50.000–79.000	59.000	44.000–69.000	0.03524 *
NK cells	14.780	4.160–24.930	13.225	4.900–23.140	0.17752
Lymphocytes T CD3+	70.540	57.270–84.200	76.110	63.340–87.660	0.00157 *
Lymphocytes B CD19+	14.170	7.110–22.660	9.940	4.350–22.120	0.00185 *
Lymphocytes T CD4+ CD3+	43.370	29.910–63.960	48.285	38.970–61.220	0.00196 *
Lymphocytes T CD8+ CD3+	25.150	16.660–41.760	25.615	20.580–36.710	0.97410
Ratio of lymphocytes T CD4+ CD3+ to CD8+ CD3+	1.692	0.716–3.839	1.858	1.125–2.688	0.42480
Lymphocytes T CD3+ CD69+	4.970	1.400–12.320	2.610	1.770–5.740	0.00070 *
Lymphocytes B CD19+ CD69+	5.890	1320–15,420	6755	2170–15,400	0.50109
Lymphocytes T CD3+ CD25+	45.590	12.970–89.930	30.015	23.460–51.120	0.00010 *
Lymphocytes B CD19+ CD25+	31.520	10.000–74.640	25.275	11.860–45.480	0.32289
Lymphocytes T CD4+ CD69+	4.700	1.230–13.840	2.135	1.170–7.580	0.00153 *
Lymphocytes T CD8+ CD69+	3.700	1.060–10.990	1.095	0.290–4.190	<0.00001 *
Lymphocytes T CD4+ CD25+	58.000	29.610–96.690	45.095	29.980–67.740	0.00698 *
Lymphocytes T CD8+ CD25+	5.960	1.450–50.410	1.760	0.550–4.590	0.00241 *
Lymphocytes B CD19+ PD-1	4.150	1.140–10.870	7.105	1.730–11.340	0.00030 *
Lymphocytes T CD4+ PD-1	20.740	12.210–32.160	7.800	5.090–10.240	<0.00001 *
Lymphocytes T CD8+ PD-1	17.450	10.120–32.450	4.015	1.320–8.360	<0.00001 *

* Statistically significant difference.

**Table 3 cancers-14-00480-t003:** Characterization of the immunophenotype tumor and lymph node samples in patients diagnosed with laryngeal cancer, considering the statistical significance between all analyzed samples of biological material (ANOVA).

Frequency of Occurrence [%]	Tumor Sample (*n* = 45)		Lymph Node Sample (*n* = 45)		Blood Sample (*n* = 45)		ANOVA *p*	Statistically Significant Difference
	Median	Range	Median	Range	Median	Range		
Lymphocytes T CD3+ CD69+	38.920	5.630–74.510	25.740	2.740–76.580	4.970	1.400–12.320	<0.00001	a–c
Lymphocytes B CD19+ CD69+	22.350	10.870–56.670	25.410	10.230–65.150	5.890	1.320–15.420	<0.00001	b,c
Lymphocytes T CD3+ CD25+	28.950	9.250–67.180	27.970	7.360–51.720	45.590	12.970–89.930	<0.00001	b,c
Lymphocytes B CD19+ CD25+	20.450	4.670–81.110	22.210	12.210–39.920	31.520	10.000–74.640	0.03831	c
Lymphocytes T CD4+ CD69+	45.450	15.670–81.510	13.270	2.850–38.590	4.700	1.230–13.840	<0.00001	a–c
Lymphocytes T CD8+ CD69+	38.890	11.250–89.920	15.580	2.390–47.970	3.700	1.060–10.990	0.00001	a–c
Lymphocytes T CD4+ CD25+	27.460	6.570–64.140	32.140	10.230–60.470	58.000	29.610–96.690	<0.00001	b,c
Lymphocytes T CD8+ CD25+	8.780	2.120–21.120	6.410	1.960–34.160	5.960	1.450–50.410	0.04731	a
Lymphocytes B CD19+ PD-1	22.470	12.250–50.250	9.210	3.470–22.140	4.150	1.140–10.870	<0.00001	a–c
Lymphocytes T CD4+ PD-1	37.140	15.890–75.450	21.120	9.180–42.870	20.740	12.210–32.160	<0.00001	a,b
Lymphocytes T CD8+ PD-1	40.410	10.450–79.450	20.870	7.120–32.460	17.450	10.120–32.450	<0.00001	a,b

(a) Statistically significant difference (*p* < 0.05) between tumor and lymph node parameters; (b) statistically significant difference (*p* < 0.05) between tumor and blood parameters; (c) statistically significant difference (*p* < 0.05) between lymph and blood node parameters.

**Table 4 cancers-14-00480-t004:** Relationships among blood, tumor, and lymph node PD-1 expression and grade/stage of the disease (ANOVA).

Characteristic Parameter	G = 1		G = 2		G = 3		ANOVA *p*	Statistically Significant Difference
Blood	Median	Range	Median	Range	Median	Range		
Lymphocytes B CD19+ PD-1	3.435	1.140–6.610	4.690	1.890–10.870	4.250	1.490–6.470	0.2060	NS
Lymphocytes T CD4+ PD-1	20.925	15.470–30.270	20.120	12.350–30.060	21.670	12.210–32.160	0.6110	NS
Lymphocytes T CD8+ PD-1	16.500	11.670–29.470	14.670	10.120–19.450	24.120	15.010–32.450	<0.0001 *	b,c
Tumor								
Lymphocytes B CD19+ PD-1	22.425	12.250–47.150	27.120	15.460–50.250	21.070	12.450–38.740	0.3480	NS
Lymphocytes T CD4+ PD-1	23.340	15.890–29.140	36.170	31.020–48.970	58.670	49.450–75.450	<0.0001 *	a–c
Lymphocytes T CD8+ PD-1	34.220	10.450–49.680	32.140	21.120–70.170	66.7950	29.460–79.450	<0.0001*	b,c
Lymph node								
Lymphocytes B CD19+ PD-1	9.530	3.470–22.140	6.570	3.470–14.460	10.520	6.150–21.120	0.0680	NS
Lymphocytes T CD4+ PD-1	15.735	11.360–24.740	21.120	9.180–37.460	31.9050	11.970–42.870	<0.0001 *	b,c
Lymphocytes T CD8+ PD-1	15.005	7.120–28.190	20.010	7.150–29.640	25.2450	22.120–32.460	<0.0001 *	b,c

NS: Not significant; * Statistically significant difference; (a) Statistically significant difference (*p* < 0.05) between groups G = 1 and G = 2; (b) statistically significant difference (*p* < 0.05) between groups G = 1 and G = 3; (c) statistically significant difference (*p* < 0.05) between groups G = 2 and G = 3.

**Table 5 cancers-14-00480-t005:** Relationships among blood, tumor, and lymph node PD-1 expression with the size of the tumor (T) based on the TNM scale of the laryngeal cancer classification (ANOVA).

Characteristic Parameter	T = 2		T = 3		T = 4		ANOVA *p*	Statistically Significant Difference
Blood	Median	Range	Median	Range	Median	Range		
Lymphocytes B CD19+ PD-1	5.250	3.110–7.180	4.250	2.210–10.870	3.680	1.140–8.170	0.3568	NS
Lymphocytes T CD4+ PD-1	20.120	19.470–22.120	17.490	12.350–30.270	22.120	12.210–32.160	0.5243	NS
Lymphocytes T CD8+ PD-1	10.120	10.120–11.450	14.240	11.670–15.320	19.450	15.670–32.450	0.0000 *	a,b
Tumor								
Lymphocytes B CD19+ PD-1	32.750	22.140–50.250	23.240	12.750–44.160	22.140	12.250–47.150	0.3092	NS
Lymphocytes T CD4+ PD-1	36.790	32.150–38.740	31.750	15.890–67.480	40.230	17.240–75.450	0.4364	NS
Lymphocytes T CD8+ PD-1	36.890	36.470–45.170	27.940	10.450–70.250	42.120	17.690–79.450	0.4193	NS
Lymph node								
Lymphocytes B CD19+ PD-1	6.140	5.460–13.780	8.180	4.310–17.640	9.470	3.470–22.140	0.6771	NS
Lymphocytes T CD4+ PD-1	16.120	10.140–16.660	24.740	11.140–37.890	21.120	9.180–42.870	0.3093	NS
Lymphocytes T CD8+ PD-1	12.690	9.340–29.640	20.120	7.120–31.460	22.110	9.780–32.460	0.5419	NS

NS: Not significant; * Statistically significant difference; (a) statistically significant difference (*p* < 0.05) between groups T = 2 and T = 4; (b) statistically significant difference (*p* < 0.05) between groups T = 3 and T = 4.

**Table 6 cancers-14-00480-t006:** Relationships among blood, tumor, and lymph node PD-1 expression and the involvement of local lymph nodes (N) based on the TNM scale of the laryngeal cancer classification. (ANOVA).

Characteristic Parameter	N = 0	N = 1	N = 2	N = 3	ANOVA *p*	Statistically Significant Difference
Blood	Median (Range)	Median (Range)	Median (Range)	Median (Range)		
Lymphocytes B CD19+ PD-1	5.480 (3.680–10.870)	3.1250 (1.140–6.140)	3.950 (1.490–8.170)	5.250 (4.460–6.470)	0.0202 *	a,b,e
Lymphocytes T CD4+ PD-1	20.620 (13.310–27.450)	21.1950 (16.670–30.270)	20.5550 (12.210–32.160)	19.470 (18.720–27.150)	0.9539	NS
Lymphocytes T CD8+ PD-1	11.890 (10.120–15.670)	16.50 (10.120–17.450)	19.0750 (12.650–32.450)	29.470 (11.450–31.170)	0.0035 *	b–e
Tumor						
Lymphocytes B CD19+ PD-1	37.310 (32.750–46.660)	21.6150 (12.250–34.120)	22.810 (12.250–44.160)	38.740 (21.020–50.250)	0.0025 *	a,b
Lymphocytes T CD4+ PD-1	31.6750 (15.890–40.020)	30.080 (17.240–48.970)	44.840 (20.670–75.450)	67.270 (22.120–72.680)	0.0104 *	c–e
Lymphocytes T CD8+ PD-1	30.140 (10.450–36.470)	31.8750 (17.690–57.890)	45.440 (21.120–79.450)	55.460 (36.890–70.290)	0.0206 *	b,c
Lymph node						
Lymphocytes B CD19+ PD-1	7.360 (6.140–12.210)	6.870 (5.460–22.140)	10.520 (3.470–21.140)	9.470 (8.180–16.780)	0.8375	NS
Lymphocytes T CD4+ PD-1	14.9450 (11.140–16.660)	16.8950 (9.180–25.340)	23.430 (11.250–42.870)	34.620 (10.140–39.460)	0.0267 *	b–e
Lymphocytes T CD8+ PD-1	8.2450 (7.120–9.780)	12.340 (10.120–17.560)	22.1150 (17.690–27.650)	31.460 (28.190–32.460)	<0.0001 *	b–f

NS: Not significant; * Statistically significant difference; (a) Statistically significant difference (*p* < 0.05) between groups N = 0 and N = 1; (b) statistically significant difference (*p* < 0.05) between groups N = 0 and N = 2; (c) statistically significant difference (*p* < 0.05) between groups N = 0 and N = 3; (d) statistically significant difference (*p* < 0.05) between groups N = 1 and N = 2; (e) statistically significant difference (*p* < 0.05) between groups N = 1 and N = 3; (f) statistically significant difference (*p* < 0.05) between groups N = 2 and N = 3.

**Table 7 cancers-14-00480-t007:** Relationships among blood, tumor, and lymph node PD-1 expression with the occurrence of metastases in distant organs of patients with laryngeal cancer (M) based on the TNM scale of the laryngeal cancer classification. (ANOVA).

Characteristic Parameter	M = 0		M = 1		*p*
Blood	Median	Range	Median	Range	
Lymphocytes B CD19+ PD-1	3.965	1.140–10.870	4.250	2.110–6.470	0.7517
Lymphocytes T CD4+ PD-1	20.005	12.210–30.270	25.150	15.670–32.160	0.1238
Lymphocytes T CD8+ PD-1	17.185	10.120–29.470	27.450	14.470–32.450	0.0008 *
Tumor					
Lymphocytes B CD19+ PD-1	21.960	12.250–50.250	31.170	21.020–38.740	0.2723
Lymphocytes T CD4+ PD-1	32.655	15.890–60.870	68.900	33.250–75.450	<0.0001 *
Lymphocytes T CD8+ PD-1	35.310	10.450–67.450	70.170	55.460–79.450	<0.0001 *
Lymph node					
Lymphocytes B CD19+ PD-1	10.070	3.470–22.140	8.210	5.640–16.780	0.3755
Lymphocytes T CD4+ PD-1	16.990	9.180–32.150	37.150	27.460–42.870	<0.0001 *
Lymphocytes T CD8+ PD-1	19.675	7.120–29.640	27.120	20.870–32.460	0.0002 *

* Statistically significant difference.

**Table 8 cancers-14-00480-t008:** The results of the analysis of the Cox proportional hazard regression model in patients with laryngeal cancer.

Type of Sample	Frequency of Occurrence [%]	The Cox Proportional Hazard Regression Model Chi-Squared: 64.60644; df = 9; *p* = 0.00000			
		*p*	Relative Hazard	Relative Hazard 95% Lower	Hazard Relative 95% Upper
Blood	Lymphocytes B CD19+ PD-1	0.060	0.516	0.259	1.028
	Lymphocytes T CD4+ PD-1	0.374	1.063	0.929	1.218
	Lymphocytes T CD8+ PD-1	0.438	0.944	0.815	1.093
Tumor	Lymphocytes B CD19+ PD-1	0.297	0.942	0.843	1.054
	Lymphocytes T CD4+ PD-1	0.994	1.000	0.921	1.086
	Lymphocytes T CD8+ PD-1	0.853	0.995	0.939	1.053
Lymph node	Lymphocytes B CD19+ PD-1	0.180	0.859	0.688	1.073
	Lymphocytes T CD4+ PD-1	0.000 *	1.480	1.189	1.842
	Lymphocytes T CD8+ PD-1	0.061	1.352	0.986	1.854

* Statistically significant difference.

**Table 9 cancers-14-00480-t009:** Analysis of the relationship between the presence of the EBV virus and the amount of IgM and IgG EBV in the blood, tumor, and lymph nodes, as well as the G grade (stage of the disease).

Characteristic Parameter	G = 1		G = 2		G = 3		ANOV *p*	Statistically Significant Difference
	Median	Range	Median	Range	Median	Range		
EBV DNA copy number/µg DNA in the tumor tissue	514.117	213.203–604.152	367.46	112.884–715.071	942.979	75.762–1542.67	0.1100	NS
EBV DNA copy number/µg DNA in the lymph node	376.585	127.357–454.85	232.516	78.20–473.254	810.267	59.894–867.907	0.0470 *	a
EBV DNA copy number/µg DNA in the blood	159.315	83.984–269.63	105.616	43.873–305.320	445.997	32.471–718.919	0.0220 *	a
IgM VCA EBV	20.120	16.070–49.910	16.540	15.310–30.000	29.055	19.510–49.740	0.0611	NS
IgG VCA EBV	95.40	27.860–103.410	64.420	37.150–135.610	77.035	31.560–148.30	0.9771	NS

NS: Not significant; * Statistically significant difference; (a) Statistically significant difference (*p* < 0.05) between groups G = 2 and G = 3.

**Table 10 cancers-14-00480-t010:** Analysis of the relationship between the presence of the EBV virus and the amount of IgM and IgG EBV in the blood, tumor, and lymph nodes with the size of the tumor (T) based on the TNM scale of the laryngeal cancer classification.

Characteristic Parameter	T = 2		T = 3		T = 4		ANOV *p*	Statistically Significant Difference
	Median	Range	Median	Range	Median	Range		
EBV DNA copy number/µg DNA in the tumor tissue	160.807	112.884–208.729	202.499	75.762–499.238	715.91	102.872–1542.67	0.0024 *	NS
EBV DNA copy number/µg DNA in the lymph node	101.686	78.2–125.171	154.125	59.894–361.409	519.145	70.562–867.907	0.0013 *	a
EBV DNA copy number/µg DNA in the blood	53.878	43.873–63.884	99.838	32.471–146.741	357.678	52.55–718.919	0.0062 *	a
IgM VCA EBV	16.135	15.4–16.87	24.825	16.54–40.3	24.47	15.31–49.91	0.4190	NS
IgG VCA EBV	50.785	37.15–64.42	80.38	37.18–100.12	77.035	27.86–148.3	0.4370	NS

NS: not significant; * Statistically significant difference; (a) statistically significant difference (*p* < 0.05) between groups T = 3 and T = 4.

**Table 11 cancers-14-00480-t011:** Analysis of the relationship between the presence of the EBV virus and the amount of IgM and IgG EBV in the blood, tumor, and lymph nodes, as well as the involvement of local lymph nodes (N) based on the TNM scale of the laryngeal cancer classification.

Characteristic Parameter	N = 0	N = 1	N = 2	N = 3	ANOV *p*	Statistically Significant Difference
	Median Range	Median Range	Median Range	Median Range		
EBV DNA copy number/µg DNA in the tumor tissue	208.729 208.729–208.729	451.782 112.884–715.071	556.572 102.872–1542.67	800.029 75.762–1000.468	0.6480	NS
EBV DNA copy number/µg DNA in the lymph node	125.171 125.171–125.171	347.553 78.2–473.254	379.745 70.562–867.907	651.149 59.894–858.16	0.4790	NS
EBV DNA copy number/µg DNA in the blood	63.884 63.884–63.884	122.811 43.873–305.32	162.658 52.55–718.919	364.969 32.471–504.924	0.5032	NS
IgM VCA EBV	16.87 16.87–16.87	15.57 15.31–30	24.47 16.07–32.2	45.02 22.85–49.91	0.0011 *	a,b
IgG VCA EBV	37.15 37.15–37.15	72.65 59.22–135.61	76.59 31.56–148.3	77.035 27.86–88.11	0.6233	NS

NS: Not significant; * Statistically significant difference; (a) Statistically significant difference (*p* < 0.05) between groups N = 1 and N = 3; (b) statistically significant difference (*p* < 0.05) between groups N = 2 and N = 3.

**Table 12 cancers-14-00480-t012:** Analysis of the relationship between the presence of the EBV virus and the amount of IgM and IgG EBV in the blood, tumor, and lymph nodes with occurrence of metastases in distant organs of patients (M) based on the TNM scale of the laryngeal cancer classification.

Characteristic Parameter	M = 0		M = 1		*p*
	Median	Range	Median	Range	
EBV DNA copy number/µg DNA in the tumor tissue	514.117	112.884–1542.67	716.748	75.762–1212.282	0.8446
EBV DNA copy number/µg DNA in the lymph node	376.585	78.2–867.907	565.037	59.894–863.076	0.4596
EBV DNA copy number/µg DNA in the blood	159.315	43.873–718.919	410.037	32.471–675.295	0.3182
IgM VCA EBV	18.940	15.31–49.910	28.830	16.54–49.740	0.14942
IgG VCA EBV	72.650	27.86–135.610	73.790	31.56–148.30	0.94265

**Table 13 cancers-14-00480-t013:** Correlations between the presence of EBV genetic material and the level of PD-1 expression in the tested biological materials from patients diagnosed with laryngeal cancer.

Type of Sample	Frequency of Occurrence [%]	EBV DNA Copy Number/µg DNA in the Blood	EBV DNA Copy Number/µg DNA in the Tumor Tissue	EBV DNA Copy Number/µg DNA in the Lymph Node
Blood	Lymphocytes B CD19+ PD-1	−0.0489 *p* = 0.762	−0.0462 *p* = 0.774	−0.0043 *p* = 0.979
	Lymphocytes T CD4+ PD-1	0.2952 *p* = 0.061	0.3385 *p* = 0.030 *	0.3705 *p* = 0.0017 *
	Lymphocytes T CD8+ PD-1	0.6969 *p* = 0.000 *	0.6282 *p* = 0.000 *	0.6760 *p* = 0.000 *
Tumor	Lymphocytes B CD19+ PD-1	0.1275 *p* = 0.427	0.0791 *p* = 0.623	0.1155 *p* = 0.472
	Lymphocytes T CD4+ PD-1	0.5289 *p* = 0.000 *	0.4304 *p* = 0.005 *	0.4850 *p* = 0.001 *
	Lymphocytes T CD8+ PD-1	0.4469 *p* = 0.003 *	0.3727 *p* = 0.016 *	0.3902 *p* = 0.012 *
Lymph node	Lymphocytes B CD19+ PD-1	0.0496 *p* = 0.758	0.0870 *p* = 0.589	0.0852 *p* = 0.596
	Lymphocytes T CD4+ PD-1	0.5461 *p* = 0.000 *	0.4523 *p* = 0.003 *	0.4817 *p* = 0.001 *
	Lymphocytes T CD8+ PD-1	0.4173 *p* = 0.007 *	0.3604 *p* = 0.021	0.4193 *p* = 0.006 *

* Statistically significant difference.

## Data Availability

The data presented in this study are available on request from the first Author (J.K.)

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
