# Peer review of "The Clinical, Pathological, and Prognostic Value of High PD-1 Expression and the Presence of Epstein–Barr Virus Reactivation in Patients with Laryngeal Cancer"

_cancers, 2022, doi:10.3390/cancers14030480_

Round 1

Reviewer 1 Report

Tha aim of the work is really interesting, but the hypothesis, the tests employed and mainly the conclusions are too confusing. I think the background must be shortened aiming to a better understanding by the reader. At the same way the discussion paragraph must be more coincided and much more short, because it is not clear the real scope of the work. Finally, the authors should clearly indicate what is the purpose of the work, if is there an hypothesis and mainly, what are the main conclusions of the work, because the "discussion" such as the background are too large

Author Response

Dear Reviewer,

Thank you very much for your valuable comments in order to improve this manuscript. Following your recommendations, we have re-edited and shortened the Introduction section and the Discussion section. In addition, we specified the purpose of our research and concluded the main conclusions of the work in the conclusion section.

We hope that in its current form the paper may be accepted, thank you.

Kind regards,

On behalf of the Authors,

Paulina Niedźwiedzka-Rystwej

Reviewer 2 Report

The study is interesting, but I don't understand if it is really important for the patients. It should be much more described and highlighted and explained why this research of the different cell populations and subpopulations in the tumor, lymph node and blood in patients with laryngeal cancer is so important. What does this mean in the end? Is it a diagnostic tool? Somehow this falls short for me.

In line 121 it says:

The control group were anti- VCA IgM negative and anti- VCA IgG positive. They were aged between

49 to 80 years, with an average age of 60,3 ± 7,63 years.

However in line 204 it says:

The control group consisted of 20 age-matched subjects (mean age 58,450 ± 7.03 years).

Why are there two different control groups?

Author Response

Dear Reviewer,

Thank you very much for your valuable comments in order to improve this manuscript. From the point of view of patients diagnosed with laryngeal cancer, it seems necessary to study the factors influencing disease progression and the occurrence of metastases to local lymph nodes and other organs. The demonstration of the presence of EBV I genetic material and its positive correlation with the expression of the PD-1 receptor in patients with laryngeal cancer may indicate a disturbance of the immune homeostasis of the organism and antitumor defense. A significantly higher percentage of T and B lymphocytes positive for PD-1 antigen expression in the laryngeal cancer tissue compared to their percentage in the blood and lymph nodes indicates that the neoplastic tissue is a place where defense mechanisms are particularly weakened. The research results obtained by our team may suggest the use of a potential anticancer therapy targeting the PD-1 / PD-L1 pathway. Additionally, our analysis of Cox's proportional hazards suggests that the assessment of PD-1 expression on TCD4 + lymphocytes located in the lymph nodes may be of prognostic significance, which may improve the diagnostic process in the future.

In line 121 it says:

The control group were anti- VCA IgM negative and anti- VCA IgG positive. They were aged between 49 to 80 years, with an average age of 60,3 ± 7,63 years.

However in line 204 it says:

The control group consisted of 20 age-matched subjects (mean age 58,450 ± 7.03 years).

Why are there two different control groups? 

Thank you very much for this apt remark. Only one control group was used in the study and there was a bug in section 2.1 that has already been corrected by us: “They were aged between 44 to 69 years, with an average age of 58.45 ± 7.03 years.”

We do hope that after the corrections, the manuscript will fulfill the requirements.

Kind regards,

Paulina Niedźwiedzka-Rystwej

Reviewer 3 Report

Dear authors

I have read your manuscript entitled "The Clinical, Pathological and Prognostic Value of High PD-1 Expression and the Presence of Epstein-Barr Virus Reactivation in Patients with Laryngeal Cancer".

Although this research is novel, i have some serious concerns regarding the manuscript:

MAJOR CONCERNS

  • The English in the article needs substantial editing as in its current form, the article is difficult to read.
  • No clear introduction is given. The authors provide numerous hypotheses on the correlation between EBV and the immunological state of laryngeal carcinoma. This makes it difficult for the reader to understand the specific approach of the authors. Therefore, I advice to rewrite the introduction section to clearly depict the current state-of-the art followed by the objective of the current research. 
  • The section regarding the statistical analysis should be adapted. F.i., you mention that data is expressed as mean +/- SD. However, in table 2 and 3, also median and range are given. Next, you state the use of ANOVA and student-t tests. This depends entirely on the  normal distribution of each parameter. If the distribution of a parameter is not-normal, one must use a Kruskal-Wallis or Mann-Whitney U test. This should be covered in the statistical analysis section.
  • Most tables is difficult to interpret.
    • Table 2: only present median values and range. This as the median value will correspond to the mean value in case of normally distributed parameters. Also only provide p-value for the results (t(63) has no additional value).
    • Table 3:  only present median values and range. This as the median value will correspond to the mean value in case of normally distributed parameters. Also add the result for blood, as you will also compare this with the tumor and lymph node samples (more clearly for readers). Only provide a single p-value (from the ANOVA / Kruskal Wallis test) for the results (t has no additional value). In case of significance, more details can be provided using symbols with a respective explanation in the table legend.
    • Table 4: incomprehensible. The data provided in tables S1 - S4 are the data that need to be presented. In this matter, it would be more appropiate to only refer to these tables and to omit Table 4 from the manuscript. Layout of Tables S1 - S4 should be parallel to the suggestions for Table 3.
    • Table 5: please omit t and Wald as these do not provide any added value. Also indicate how the hazard should be interpret (e.g. per increasing % ???). This also needs to be specified in the text (line 331 - 333).
    • Table 6: same remarks as for Table 4 (only refer to Tables S6 - S9 and adapt layout). Also omit the descriptive statistics and replace them to the main text (e.g. at line 348 - 350). 
  • Both figures are redundant and provide no additional value
    • Figure 1: blurry figure. It would be more meaningful to provide a flow cytometric figure which allows for visual comparison of the statistically significant subpopulations (e.g. CD8+PD-1+)
    • Figure 2: totally no added value for this manuscript. Please omit.
  • Discussion: when reading the discussion section, I have noticed that you restrict yourselves to a repetition of the most significant results. I do however also miss some actual conclusion drawing, as well as formation of novel research hypotheses on why you have noticed certain changes. Moreover, I was suppressed that you did not include any study limitations (e.g. small sample size). This is of the utmost importance and should be provided.

MINOR CONCERNS

  • No uniform use of abbreviations throughout the manuscript (e.g. PBS). Please adapt.
  • Line 57 - 58: data show that in 2020 19,63 people in Europe died of laryngeal cancer (which accounted for 19.63% of all deaths in the world), > 19,63 people seems incorrect
  • Line 95: it is not the PD-1 receptor, but the PD-1 / PD-L1 complex that inhibits the autoimmune response. Please adapt.
  • Line 115: Please adapt to “Forty-five untreated patients…”
  • Line 287 - 290: remove as this belongs in the introduction.
  • Line 331 – 333: did you attempt to determine the prognostic value of CD4+PD-1+ lymphocytes from the lymph node as a categorical variable? As this would be more meaningful in a clinical setting in comparison to a continuous variable.

Author Response

Dear Reviewer,

Thank you very much for all the valuable comments that will surely help our manuscript to improve. We tried to correct and answer all of your concerns and here is the point-by-point answer to your suggestions:

MAJOR CONCERNS

  • The English in the article needs substantial editing as in its current form, the article is difficult to read.

The language has been redrafted in the publication, which we hope will make it easier for readers to choose.

  • No clear introduction is given. The authors provide numerous hypotheses on the correlation between EBV and the immunological state of laryngeal carcinoma. This makes it difficult for the reader to understand the specific approach of the authors. Therefore, I advice to rewrite the introduction section to clearly depict the current state-of-the art followed by the objective of the current research. 

In line with your recommendations, we have re-edited part of the introduction to clearly present the current state of knowledge and modified the paragraph on the purpose of the research to make it clear our point of view.

  • The section regarding the statistical analysis should be adapted. F.i., you mention that data is expressed as mean +/- SD. However, in table 2 and 3, also median and range are given. Next, you state the use of ANOVA and student-t tests. This depends entirely on the  normal distribution of each parameter. If the distribution of a parameter is not-normal, one must use a Kruskal-Wallis or Mann-Whitney U test. This should be covered in the statistical analysis section.

In line with your recommendations, we have detailed the scope of the statistical analyzes performed, which are discussed in section 2.6:

„The experimental continuous data were expressed as mean ± SD,  median and range, and analyzed by Statistica 12 software (StatSoft, USA). Several types of statistical analyses were performed. Normality of data distribution was tested with the Shapiro-Wilk test. Levene's test was used to evaluate the homogeneity of variance. Student’s t-tests either independent or paired were used when comparing normally distributed variables, and Mann-Whitney U test or Wilcoxon signed rank test in case of non-normally distributed variables. The differences between more than two groups were analyzed by ANOVA or ANOVA Kruskal-Wallis. The Pearson correlation coefficients and corresponding significance tests were used to assess the strength and direction of the linear relationships between pairs of variables. The Cox proportional hazard regression model was also evaluated. A value p less than 0.05 was considered statistically significant.”

  • Most tables is difficult to interpret.
    • Table 2: only present median values and range. This as the median value will correspond to the mean value in case of normally distributed parameters. Also only provide p-value for the results (t(63) has no additional value).

The table has been redrafted as suggested, including medians, range and p-value to make it easier to read.

    • Table 3:  only present median values and range. This as the median value will correspond to the mean value in case of normally distributed parameters. Also add the result for blood, as you will also compare this with the tumor and lymph node samples (more clearly for readers). Only provide a single p-value (from the ANOVA / Kruskal Wallis test) for the results (t has no additional value). In case of significance, more details can be provided using symbols with a respective explanation in the table legend.

The table has been revised according to the guidelines. A column for the median value and range has been added for the blood samples. Statistically significant differences are presented using letter abbreviations, the detailed characteristics of which are in the description under the table.

    • Table 4: incomprehensible. The data provided in tables S1 - S4 are the data that need to be presented. In this matter, it would be more appropiate to only refer to these tables and to omit Table 4 from the manuscript. Layout of Tables S1 - S4 should be parallel to the suggestions for Table 3.

Data from Table 4 and Tables from S1-S4 supersets were used to create four independent tables, presenting: Relationships between blood, tumor and lymph node PD 1 expression and grade G and TNM scale. Each of the tables corresponding to the individual G, T, N, M parameters contains data on medians, range and statistical significance expressed as letter symbols, the legend of which is placed under the table.

    • Table 5: please omit t and Wald as these do not provide any added value. Also indicate how the hazard should be interpret (e.g. per increasing % ???). This also needs to be specified in the text (line 331 - 333).

Table 5. now Table 8. Has been revised as suggested.

    • Table 6: same remarks as for Table 4 (only refer to Tables S6 - S9 and adapt layout). Also omit the descriptive statistics and replace them to the main text (e.g. at line 348 - 350). 

Data from Table 4 (now Table 9) and Tables from S6-S9 supplements were used to create four independent tables, presenting: Analysis of the relationship between the presence of the EBV virus and the amount of IgM and IgG EBV in the blood, tumor and lymph nodes, and the G grade and TNM scale. Each of the tables corresponding to the individual G, T, N, M parameters contains data on medians, range and statistical significance expressed as letter symbols, the legend of which is placed under the table.

  • Both figures are redundant and provide no additional value
    • Figure 1: blurry figure. It would be more meaningful to provide a flow cytometric figure which allows for visual comparison of the statistically significant subpopulations (e.g. CD8+PD-1+)

Figures showing cytometric analysis have been added:

Figure 1. Example of the cytometric analysis of CD19+PD-1+ lymphocytes in blood sample (A), tumor sample (B) and lymph node sample (C).

Figure 2. Example of the cytometric analysis of CD4+PD-1+ lymphocytes in blood sample (A), tumor sample (B) and lymph node sample (C).

Figure 3. Example of the cytometric analysis of CD8+PD-1+ lymphocytes in blood sample (A), tumor sample (B) and lymph node sample (C).

    • Figure 2: totally no added value for this manuscript. Please omit.

The second drawing has been removed from work

  • Discussion: when reading the discussion section, I have noticed that you restrict yourselves to a repetition of the most significant results. I do however also miss some actual conclusion drawing, as well as formation of novel research hypotheses on why you have noticed certain changes. Moreover, I was suppressed that you did not include any study limitations (e.g. small sample size). This is of the utmost importance and should be provided.

In line with your recommendations, we edited and shortened the discussion as suggested, and added a paragraph on the limitations of the research conducted.

4.4 Limitations of conducted research

Despite obtaining statistically significant results, our research has some kind of limitations. One of the limitations of our research team is undoubtedly the small sample size of the tested patients diagnosed with laryngeal cancer. Some of the patients recruited for the study do not reflect the nature of the entire population of people affected by this disease, which may also affect the results of our research. Additionally, only male patients participated in the study, which limits the results obtained only to this group of patients. The strength of our research is a comprehensive analysis of both PD-1 receptor expression and the presence of EBV in three types of biological material obtained from patients (peripheral blood, tumor samples and lymph node) and taking into account the interpretation of the results based on the severity of the disease and the TNM scale. The obtained results provide information on the contribution of the analyzed parameters to the development and progression of laryngeal cancer. However, further studies are needed to clarify the exact role of the PD-1 receptor and the presence of EBV in the pathogenesis of laryngeal cancer, and to extend the scope of studies to a larger number of patients (including female participation). Thanks to this, it will be possible to determine whether the percentage of PD-1 receptors on TCD4 + lymphocytes selected in these studies will have a prognostic value, allowing the inclusion of this parameter in the diagnostic panel for laryngeal cancer.

MINOR CONCERNS

  • No uniform use of abbreviations throughout the manuscript (e.g. PBS). Please adapt.

We standardized the abbreviations throughout the manuscript

  • Line 57 - 58: data show that in 2020 19,63 people in Europe died of laryngeal cancer (which accounted for 19.63% of all deaths in the world), > 19,63 people seems incorrect

The statistical data has been checked and corrected again:

“This also applies to the number of deaths, data show that in 2020 1960419,63 people in Europe died of laryngeal cancer (which accounted for 19.63% of all deaths in the world), 89.87% of whom were men [1].”

  • Line 95: it is not the PD-1 receptor, but the PD-1 / PD-L1 complex that inhibits the autoimmune response. Please adapt.

We have implemented the suggestion

“The PD-1/PD-L1 complex that inhibits autoimmune responses by blocking T cell proliferation and cytokine production [23].”

  • Line 115: Please adapt to “Forty-five untreated patients…”

We have implemented the suggestion

 “The forty-five untreated patients with laryngeal squamous cell carcinoma (LSCC) were enrolled in Department of Otolaryngology and Oncology of the Medical University of Lublin.” 

  • Line 287 - 290: remove as this belongs in the introduction.

Since we are having difficulties with understanding the suggestion (thoe lineas are as we can see empty before the Table). Could we please ask the Reviewer to explain in a more detailed manner. We will be of course willing to correct.

  • Line 331 – 333: did you attempt to determine the prognostic value of CD4+PD-1+ lymphocytes from the lymph node as a categorical variable? As this would be more meaningful in a clinical setting in comparison to a continuous variable.

Thank you for this valuable notice, but this was the first study with a lot of limitations and we were not attempting to create a categorical variable. In our opinion, a fix number of possible values need a larger number of patients, so we would be willing to perform it in the future study.

Again, we would like to thank you for your time and consideration and we hope that the corrected manuscript will fulfill the requirements. We are eager to provide any further changes required.

Kind regards,

Paulina Niedźwiedzka-Rystwej

Round 2

Reviewer 3 Report

Dear authors

Many thanks for the substantial revision of the manuscript.

Herewith, I provide with some minor comments:

  • - All tables are more clearly now. Minor suggestion: I would round all numbers to 1 or 2 decimal points (except the P-values) in Tables 2, 3, 4, 5, 6, 7, 9, 10, 11 and 12.

  • - All figures are now more illustrative of the performed research. Minor suggestion: It would be more meaningful to link this to the results in Table 3. In this way, the reader is informed to the fact that this concerns this respective statistical difference. I would also suggest to modify the Figure legend. ”Example” gives the impression that a random plot was taken. In this case, the depiction of a plot corresponding to the median value for the respective blood, tumor and lymph node sample is advised.

  • - With regard to my previous comment: “Line 287 - 290: remove as this belongs in the introduction.” What I actually meant is that the sentence “(where T is the size of the tumor, N is the involvement of local lymph nodes, and M is the presence of metastases to distant organs)”, which can be found at the beginning of section 3.3 (page 11, line 288-289 of the revised manuscript), does not belongs to the Results section. This should be added to the introduction section, or to the Materials and Methods section were you describe the TNM scale for the first time.

Author Response

Dear Reviewer,

Thank you very much for your comments. We have followed your additional comments and left the tracking changes in the manuscript. We hope that now the manuscript will be clearer to read and will be suitable to the journal. We would like to deeply thank you for the effort and mindfulness invested in our paper. We are open to any further suggestions you may have.

Kind regards,

Paulina Niedźwiedzka-Rystwej